# Correlation of serum screening and neuroimaging studies: A comparative analysis for diagnostic evaluation of neurocysticercosis among people with epilepsy attending mental health clinics in selected district hospitals of Tanzania

**Charles E. Makasi** [1,2]*, **Bernard Ngowi**[3], **Andrew Kilale**[1], **Godfrey Guga**[4], **Michael J. Mahande**[5], **Johnson Mshiu**[1], **Abisai Kisinda**[6], **Blandina T. Mmbaga**[2,7]

**1** National Institute for Medical Research, Muhimbili Medical Research Centre, Dar es Salaam, Tanzania, **2** School of Medicine, KCMC University, Moshi, Tanzania, **3** Mbeya College of Health and Allied Sciences, University of Dar es Salaam, Mbeya, Tanzania, **4** Haydom Global Health Research Centre, Haydom Lutheran Hospital, Mbulu, Tanzania, **5** KCMC University, Moshi, Tanzania, **6** National Institute for Medical Research, Mbeya Medical Research Centre, Mbeya, Tanzania, **7** Kilimanjaro Clinical Research Institute, Moshi, Tanzania

* charlesemakasi@gmail.com

## Abstract

### Background

Involvement of *Taenia solium* cysts in the nervous system (Neurocysticercosis -NCC), is responsible for about 30% of preventable acquired epilepsy among people with epilepsy in endemic areas. The diagnosis of NCC involves a set of criteria including immunodiagnostics and neuroimaging. The gold standard immunodiagnostics is not clearly established and the available tests are commercial and not commonly practiced in our settings. We conducted this study to establish the correlation between serum cysticercosis screening tests and neuroimaging studies in the diagnosis of neurocysticercosis among people with epilepsy attending mental health clinics in endemic areas of Tanzania.

### Methods

We evaluated the serum cysticercosis diagnostic tests through parallel tests and performed multiple analysis methods including sensitivity and specificity, positive and negative likelihood ratio (LR+, LR-). Others were positive predictive value (PPV), negative predictive value (NPV), and receiver operating characteristics (ROC) curve. These were done to establish the efficiency of serum cysticercosis tests in the diagnosis of NCC.

### Results

Our findings showed a sensitivity of 87.5%; [95%CI (76.7–98.3)] and specificity of 60.7%; [95%CI (44.8–76.7)]. The LR+ = 2.23 and LR- = 0.21. The PPV was 38.9% [95%CI

**Data availability statement:** All data files are available from the figshare repository. DOI:10.6084/m9.figshare.27936138

**Funding:** The research work was funded by the Bundesministerium für Bildung und Forschung (Federal Ministry of Education and Research) (BMBF), Germany; under the Cysticercosis Network of Sub-Saharan Africa (CYSTINET-Africa) project 01KA1618 with GIZ Contract No. 81203618 for the Ph.D. capacity-building plan. The funders had no role in study design, data collection, and analysis, decision to publish, or preparation of the manuscript

**Competing interests:** The authors have declared that no competing interests exist

(23.0–54.8)], NPV was 94.4%; [95%CI (87.0–101.9)] and area under the ROC curve was 0.78.

## Discussion

Increasing sensitivity and decreasing false negatives, parallel testing of multiple serum cysticercosis diagnostic tests enhances negative predictive values. It is important to assess the diagnostic tests using a variety of analytical techniques to confirm that they are capable of producing the desired diagnosis.

# Authors summary

Neurocysticercosis (NCC) happens when the larvae from the pork tapeworm infect the central nervous system, especially the brain. About 30% of occurrences of epilepsy that are preventable in places where this parasite is prevalent are caused by this disease. Brain scans and blood testing are typically used in the diagnosis of NCC. The most effective blood test for this is unclear, though, and our area doesn't often employ the available tests. Our goal in doing this study was to see how well brain scans and blood tests correlate for the diagnosis of NCC in people with epilepsy living in regions of Tanzania where the parasite is prevalent.

Based on our findings, combining several blood tests improve the precision of identifying individuals without NCC and lowers the possibility of false negative results. To ensure that diagnostic tests are effective in diagnosing the ailment, it's critical to assess them using several methodologies.

## Introduction

*Taenia solium (T. solium)* infection can manifest as taeniasis which is the presence of an adult tapeworm in the intestines since a human being is the only natural definitive host [1]. Cysticercosis (caused by *Cysticercus cellulosae*), is the larval stage of *T. solium* in the tissues. Pigs act as intermediate hosts of the *T. solium* larval stage, which causes porcine cysticercosis [2]. Neurocysticercosis (NCC) is when the *T. solium* cysts invade the nervous system.

Neurocysticercosis is responsible for about thirty percent of preventable acquired epilepsy, especially in endemic areas; but has been associated with as much as seventy percent in the highest-risk settings [3,4]. Areas, where free-range pig keeping is practiced, are more likely to be at a higher risk especially when personal hygiene is low and open defecation is practiced [5,6]. Studies give evidence of the presence of seventeen percent of cysticercosis seroprevalence in some areas of Tanzania and about 2 to 5 percent of the adult tapeworm in the population [7–9].

Despite *T. solium*-associated conditions like epilepsy gaining a lot of attention recently, the control of *T. solium* is not a standalone priority but is inclusively performed with other neglected diseases in Tanzania [10]. On the other hand, although NCC is responsible for about 30% of epilepsy among people with epilepsy (PWE) in highly endemic areas this cause of epilepsy is not properly evaluated in the mental health clinics of Tanzania [11]. These mental

health clinics in Tanzania provide services for different mental health disorders including epilepsy, schizophrenia, depression, bipolar disorders, and other mental health problems [12,13]. The diagnosis of NCC is complex and involves multiple approaches including epidemiological information, clinical evaluation, serological tests, and neuroimaging examination [14]. The techniques for diagnosis of NCC are available as they have been published. However, it's the availability, knowledge, and lack of equipment to perform the tests that are still challenging, especially in lower-level health facilities of developing countries [15,16].

The mode of diagnosis of NCC including neuroimaging and serological cysticercosis tests are still rarely accessible and not routinely practiced in lower and middle-income countries (LMIC) including Tanzania [17]. Serological diagnostics for cysticercosis are still commercially available in countries like Tanzania and are normally performed in advanced scientific laboratories which are very few and not evenly distributed [18]. Therefore, blood samples must be collected from the field sites and then shipped to the diagnostic points for analysis. However, the quality and reliability of a laboratory result are not based solely on the analytical process [19]. Sample collection is the first step, followed by preservation and shipping from the point of collection to the analytical point. The quality control procedures performed in the lab will be useless if the sample is delivered at an inappropriate temperature [20]. All these need to be taken into account for the credible serological diagnostic results [20]. Due to the absence of the gold standard serum cysticercosis tests, choosing an appropriate diagnostic test is sometimes challenging, especially in some local settings. Adhering to the Standards for Reporting Diagnostic Accuracy Studies (STARD) checklist; our study aims to assess the correlation between serum screening and neuroimaging studies among PWE attending mental health clinics in selected district hospitals of Tanzania. It specifically evaluates the diagnostic accuracy of the serum cysticercosis tests and compares their results with neuroimaging findings.

## 2.0. Methods

### 2.1. Ethics statement

This study was approved by The Kilimanjaro Christian Medical College Research Ethics and Review Committee (CRERC); Certificate No. 2450. The CYSTINET-Africa study proposal received ethical clearance from the National Institute for Medical Research (NIMR) with reference number NIMR/HQ/R.8a/Vol. IX/ 2529 and the Technical University of Munich, Klinikum rechts der Isar, Ethics Committee with reference number 537/18; The study was carried out in compliance with the Declaration of Helsinki. Approval to conduct this study in the selected sites was obtained after meetings with local administrative authorities to explain the study's aims and procedures. All study participants gave their written informed consent for inclusion before they participated in the study.

### 2.2. Study design and setting

A multicenter hospital-based prospective study was conducted. It involved PWE around the catchment populations of Kongwa and Chunya district hospitals in Dodoma, central Tanzania, and Mbeya Southern Tanzania. The recruited participants regularly attend mental health clinics for the care of epilepsy. The current study was a sub-study of the CYSTINET-Africa project. The study sites were the same i.e. Kongwa District Hospital at Kongwa District and Chunya District Hospital of Chunya District as described by Makasi et al., [21,22]. These two locations i.e. Chunya and Kongwa districts have similar pig-keeping practices. Despite the lack of enough data, Chunya's mining, forestry, and fishing sectors suggest a lower poverty level than the Kongwa district which has mainly subsistence farming.

## 2.3. Participants

Participants were enrolled prospectively between July 2020 and April 2021. The screening was done using an eligibility form. We recruited PWE with or without anti-seizure medication (ASM) who visited the mental health clinic, aged at least 14 years old, lived in the Kongwa and Chunya districts, consented to participate in the study, and complied with all study procedures, including the initial Computed Tomography (CT) scans and any follow-up CT scans, as well as any necessary NCC treatment. PWE with a history of substantial mental health illnesses in the past, those who were mentally incompetent and unable to follow directives, pregnant women, and those who were extremely unwell physically were all excluded. Individuals with mental health diseases such as depression or schizophrenia were also excluded.

## 2.4. Procedures

Kongwa District Hospital was one of the study sites and the recruitment took place at the mental health clinic. One clinician in charge of the mental clinic (MN) and a visiting doctor (CM) attended to potential participants. In Chunya, there was also one visiting doctor (SN) who worked for the study and the doctor in charge of the mental health clinic (DM) who recruited all study participants at this site. Patients attending the mental health clinics in Tanzania present with different problems including depression, schizophrenia, bipolar disorders, and other brain disorders like dementia. Unfortunately, the neurophysiology department is not established well in Tanzania and therefore people with epilepsy attend the same mental health clinic as other individuals with psychiatric disorders as mentioned earlier. Our study recruited PWE, hence, those without epilepsy were directed to go through the routine local standard of care. A blood sample was collected from all recruited PWE and analyzed for cysticercosis (CC) by both antigen and antibody cysticercosis tests.

During recruitment, our study enrolled people known to have epilepsy or newly diagnosed patients. Epilepsy is a neurological disorder characterized by recurrent seizures that can manifest in various ways, including convulsions, muscle spasms, loss of consciousness, and altered awareness or sensations. It is typically diagnosed clinically if a person experiences two or more unprovoked seizures separated by at least 24 hours [23,24] A convenience sampling technique was applied during recruitment, and other attendees to the mental clinic with problems other than epilepsy were not eligible for the study.

## 2.5. Data collection methods and tools

All eligible participants were asked for their consent to participate in the study and written informed consent was filled and signed. Data collection was done by well-trained medical practitioners (CM for Kongwa and SN for Chunya) who received extra training from a doctor with neurological experience. A portable electronic tablet computer installed with The KoboToolbox software program was used to collect the rest of the data required including demographic information through a standardized questionnaire. The neurological status was assessed through a detailed neurological examination, using standardized and validated forms to ensure consistency and accuracy in data collection.

## 2.6. Test methods

At each site, an assigned phlebotomist collected 20 ml of blood using a 20G syringe and 23G needle. The phlebotomist then transferred 10 ml into two tubes and each tube was pre-labeled with a barcode/unique identifier. The red top tube was left upright at room temperature for 1 hour before separating serum to ensure clotting. This ensured that a maximum amount of serum was obtained from the blood sample. The serum was collected in three cryovials; one

for testing with apDia Ag-ELISA, the second for LDBio IgG Western blot, and the third for quality control. Similar barcodes/unique were attached to the tubes, showing that all three aliquots belonged to one individual. The serum was stored at −20 °C waiting for transportation to the National Institute for Medical Research (NIMR) Mbeya laboratory.

### 2.7. Shipping of the serum samples

The distance from Chunya to the NIMR-Mbeya laboratory is about 70 km, and from Kongwa to the NIMR-Mbeya laboratory is 675 km. Recruited participants were very few in a week and it was impossible to ship the samples every day or every week from the recruiting site to the NIMR-Mbeya laboratory. Therefore, although it was suggested to ship the sample regularly possibly after every week, we had to wait until we reached 25 samples or more. Frozen samples were well labeled and prepared by trained laboratory staff, adhering to all biohazard sample regulations. They were then packed into liquid nitrogen tanks, which maintained the shipping temperature at −20 ºC or below. The samples were shipped using a project vehicle to the analytical point at the NIMR-Mbeya laboratory accompanied by a trained laboratory staff.

### 2.8. Serological diagnosis of cysticercosis

The serological tests were performed following the manufacturer's guidelines [25,26] The two serum cysticercosis tests were used in parallel. Therefore, based on our study, a case of cysticercosis was defined as a subject having tested serologically positive for cysticercosis (by either antigen or antibody or both tests).

### 2.9. Cerebral computed tomography examination

All PWE who tested positive on either of the two CC serological tests went for their bran CT scan. The CT scan examination performed in this study is the same as that described by Makasi et al., 2024 [22]. Participants from the Kongwa recruitment site were examined at the Dodoma Mbijiwe Diagnostic Centre and those from Chunya at the Mbeya Zonal Referral Hospital. Neurocysticercosis diagnosis was based on the 2017 updated Del Brutto criteria [14].

### 2.10. Diagnosis and treatment of neurocysticercosis

A neuroradiologist with NCC experience reviewed the CT images. Lesions were categorized as either parenchymal (frontal lobe, temporal lobe, parietal lobe, occipital lobe, cerebellum, brainstem) or extra-parenchymal (intraventricular or subarachnoid). Neurocysticercosis was classified as either definite or probable based on exposure, clinical, and neuroimaging criteria [14]. Treatment options for patients with NCC were agreed upon by the study clinical team which also included a neurologist.

A center for treatment of all eligible participants was established at Ifisi Hospital, Mbalizi Mbeya which is 70 km from Chunya and 675 km from Kongwa. All patients were admitted for at least 10 days and received antiparasitic (a combination of albendazole and praziquantel) treatment under very close supervision.

## 3.0. Statistical analysis

Data management and statistical analysis were performed using Stata version 17.0 (Stata Corp, Texas, USA). Descriptive statistics were used to characterize participants' sociodemographic profiles. Categorical variables were summarized using frequencies and proportions, while continuous variables were described using mean and standard deviation. Performance characteristics were analyzed, including sensitivity, specificity, positive predictive values (PPV),

negative predictive values (NPV), and their respective 95% confidence intervals. Likelihood ratio (LR) tests and ROC curves were used to assess the diagnostic performance of the predictive models. The study results were presented using tables and figures.

## 4.0. Results

### 4.1. Demographic information

A total of 510 individuals attended the mental health clinics of Kongwa and Chunya districts hospitals. We recruited a total of 223 PWE which is 43.7% of all attended individuals from both mental health clinics. Of the 223 participants recruited in our study, 25(11.2%) were CC-positive [(by either antigen (Ag) or antibody (Ab) or both)]. All the 25 PWE with CC positive were eligible for neuroimaging cerebral CT) scan examination. Twenty-eight (12.6%) of the total 223 PWE presented with obvious neurological signs and symptoms and five (3.6%) of the 28 had both neurological presentations as well as CC positive tests. Those who presented with prominent neurological signs and symptoms were also offered a CT scan examination. Only 23 (10.3%) of the 223 PWE remained with neurological presentations but CC negative. Of the 25 PWE who were CC positive, 18 (72%) of 25 PWE showed up and went for a CT scan; 7 (28%) of the 25 PWE absconded from their CT scan examination appointment. Only 18 out of 23 (78.3%) PWE CC negative and other neurological signs/symptoms positive underwent CT examination. The rest did not show up for the examination without any established reason. Altogether, 8 (22.2%) of all 36 PWE who went for CT scan; (7 from the CC-positive group and one from PWE from CC negative) with neurological presentation showed NCC-typical lesions (Fig 1).

Two hundred twenty-three people with epilepsy were recruited from the two mental clinics of Kongwa and Chunya district hospitals; 118 (52.9%) were female. The mean age of the study participants was 35.7 ± 15.0 years. Kongwa and Chunya showed a similar pattern in terms of age, sex, and religion of the participants. Kongwa recruited more than twice the number of PWE without formal education (p = 0.002) and twice the proportion of farmers (p < 0.001) compared to Chunya. Kongwa presented with more PWE with unknown HIV status compared to Chunya 116 (67.4%) vs 8 (15.7%); p < 0.001 (Table 1).

### 4.2. Number, stages, and distribution of the cysts

Of the 36 participants who were examined for NCC by the CT scan, there were no colloidal vesicular stage lesions in any of the subjects and 28 subjects, had no NCC lesions discovered in their brains. Despite varying lesion characteristics, 8 participants tested positive for NCC. The chronic character of NCC in endemic environments is consistent with the prevalence of dormant (calcified) lesions. Participants with active or mixed-stage lesions had higher rates of seroreactivity, indicating a more robust immune response (Table 2).

### 4.3. Serum cysticercosis test results and neuroimaging (Computed tomography)

At Kongwa there were CC+ Ag-tests [6/172 (3.5%) as compared to Chunya (9/51 (17.6%); p < 0.001)], but both of them had similar rates of CC+ Ab-tests [13/172 (7.6%) vs 4/51 (7.8%); P>0.9. Of all recruited PWE; 25/223 (11.2%) were CC positive on either test whereby Kongwa contributed 15/172 (8.7%) vs Chunya 10/51 (19.6%); P = **0.030.** The NCC was diagnosed among 8/36 (22.2%) with Kongwa presenting with few cases 3/26 (11.5%) vs Chunya 5/10 (50.0%); p = 0.024 irrespective of their similarities in CC+ Ab-tests (Table 3).

Of the 36 PWE, 8 (22.2%) were diagnosed as positive and 28 (77.8%) negatives for NCC by cerebral CT scan-based gold standard methods. Neuroimaging including a CT scan is so far considered the gold standard for the diagnosis of NCC and Del Brutto criteria highly depend

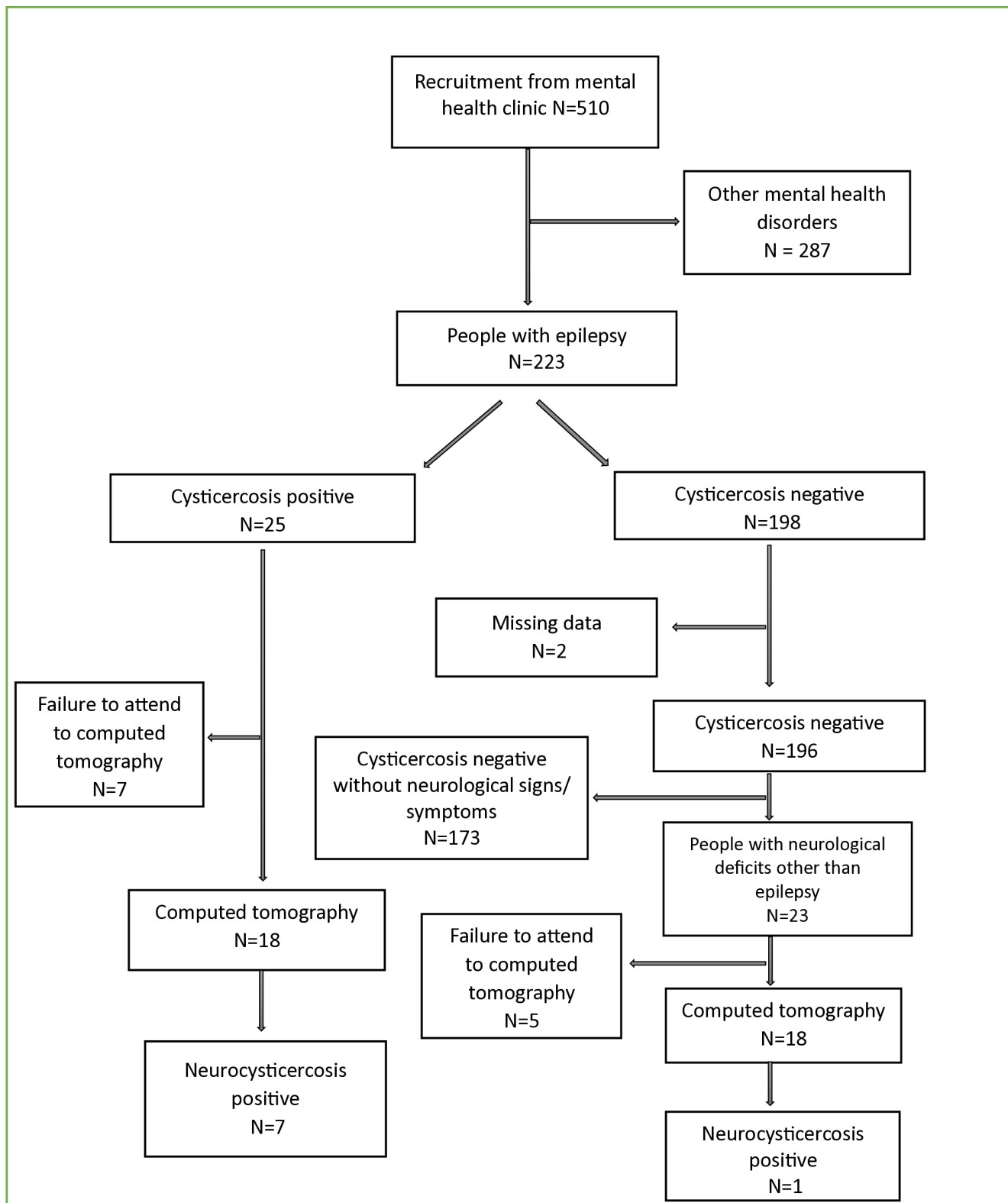

**Fig 1. Flowchart of the recruitment of people with epilepsy in the participating mental clinics.**

Table 1. Sociodemographic and clinical characteristics of people with epilepsy (N = 223).

| Characteristic | Overall | Kongwa | Chunya | p-value** |
|---|---|---|---|---|
| | n (%) | n (%) | n (%) | |
| **Total** | **223** | **172** | **51** | |
| **Sex** | | | | >0.9 |
| Female | 118 (52.9) | 91 (52.9) | 27 (52.9) | |
| Male | 105 (47.1) | 81 (47.1) | 24 (47.1) | |
| **Patients age** | | | | 0.6 |
| Mean (SD) | 35.7 (15.0) | 35.6 (15.2) | 36.1 (14.4) | |
| Median (IQR) | 32.0 (24.0, 44.5) | 31.5 (24.0, 43.2) | 36.0 (25.0, 47.5) | |
| Minimum, Maximum | 15.0, 90.0 | 18.0, 90.0 | 15.0, 65.0 | |
| **Categorical age** | | | | 0.7 |
| < 40 yrs | 151 (67.7) | 118 (68.6) | 33 (64.7) | |
| 40–59 yrs | 52 (23.3) | 38 (22.1) | 14 (27.5) | |
| 60 yrs + | 20 (9.0) | 16 (9.3%) | 4 (7.8) | |
| **Religion** | | | | 0.7 |
| Christian | 211 (94.6) | 161 (93.6) | 50 (98.0) | |
| Muslim | 9 (4.0) | 8 (4.7) | 1 (2.0) | |
| Others | 3 (1.3) | 3 (1.7) | 0 (0.0) | |
| **Education** | | | | **0.002** |
| No formal education | 77 (34.5) | 68 (39.5) | 9 (17.6) | |
| Not completed primary school | 62 (27.8) | 48 (27.9) | 14 (27.5) | |
| Primary education | 69 (30.9) | 49 (28.5) | 20 (39.2) | |
| Secondary education and above | 15 (6.7) | 7 (4.1) | 8 (15.7) | |
| **Occupation** | | | | **<0.001** |
| Farmer | 182 (82.4) | 160 (93.6) | 22 (44.0) | |
| Housewife/Student/Unemployed | 15 (6.8) | 0 (0.0) | 15 (30.0) | |
| Miners/Driver/Employed | 10 (4.5) | 2 (1.2) | 8 (16.0) | |
| Pastoralist/Pet trader | 14 (6.3) | 9 (5.3) | 5 (10.0) | |
| **Marital status** | | | | 0.14 |
| Married/Co-habiting | 84 (37.7) | 59 (34.3) | 25 (49.0) | |
| Single | 108 (48.4) | 89 (51.7) | 19 (37.3) | |
| Widowed/Divorced/Separated | 31 (13.9) | 24 (14.0) | 7 (13.7) | |
| **HIV status** | | | | <0.001 |
| HIV -ve | 97 (43.5) | 55 (32.0) | 42 (82.4) | |
| HIV +ve | 2 (0.9) | 1 (0.6) | 1 (2.0) | |
| Unknown | 124 (55.6) | 116 (67.4) | 8 (15.7) | |

** Fisher's exact test; Pearson's Chi-squared test

on the CT scan as a key diagnostic tool. Adhering to the Del Brutto Criteria 2017 consideration. (serum cysticercosis tests and cerebral CT scan positive) a total of 8 (22.2%) of all 36 cases had NCC; 7 (87.5%) of all 8 individuals who were NCC positive, were also positive for the serum cysticercosis test, and only 1 (12.5%) case of the NCC was negative for serum cysticercosis test (Table 4).

### 4.4. Sensitivity, specificity, and likelihood ratio

The sensitivity and specificity of the serum cysticercosis tests were 87.5%; [95%CI (76.7–98.3)], and 60.7%; [95%CI (44.8–76.7)], respectively. Out of 8 samples detected positive for NCC,

**Table 2. Number, stages, and distribution of the NCC lesion (N = 36).**

| ID | Vesicular stage | Colloidal vesicular stage | Granular stage | Calcified stage | Active cysts parenchymal | Active cysts extra-parenchymal | NCC (no/ inactive/ mixed/active) | NCC results |
|---|---|---|---|---|---|---|---|---|
| 1 | 0 | 0 | 0 | 0 | 0 | 0 | no | NCC -ve |
| 2 | 1 | 0 | 0 | 0 | 1 | 0 | active | NCC +ve |
| 3 | 0 | 0 | 0 | 0 | 0 | 0 | no | NCC -ve |
| 4 | 0 | 0 | 0 | 0 | 0 | 0 | no | NCC -ve |
| 5 | 0 | 0 | 0 | 0 | 0 | 0 | no | NCC -ve |
| 6 | 0 | 0 | 0 | 0 | 0 | 0 | no | NCC -ve |
| 7 | 0 | 0 | 0 | 10 | 0 | 0 | inactive | NCC +ve |
| 8 | 0 | 0 | 0 | 43 | 0 | 0 | inactive | NCC +ve |
| 9 | 0 | 0 | 0 | 76 | 0 | 0 | inactive | NCC +ve |
| 10 | 16 | 0 | 0 | 13 | 15 | 1 | active(mixed) | NCC +ve |
| 11 | 0 | 0 | 0 | 0 | 0 | 0 | no | NCC -ve |
| 12 | 0 | 0 | 0 | 0 | 0 | 0 | no | NCC -ve |
| 13 | 16 | 0 | 3 | 3 | 19 | 0 | active(mixed) | NCC +ve |
| 14 | 0 | 0 | 0 | 0 | 0 | 0 | no | NCC -ve |
| 15 | 0 | 0 | 0 | 0 | 0 | 0 | no | NCC -ve |
| 16 | 0 | 0 | 0 | 0 | 0 | 0 | no | NCC -ve |
| 17 | 0 | 0 | 0 | 0 | 0 | 0 | no | NCC -ve |
| 18 | 0 | 0 | 0 | 0 | 0 | 0 |  | NCC -ve |
| 19 | 32 | 0 | 0 | 10 | 6 | 26 | active(mixed) | NCC +ve |
| 20 | 0 | 0 | 0 | 0 | 0 | 0 | no | NCC -ve |
| 21 | 0 | 0 | 0 | 0 | 0 | 0 | no | NCC -ve |
| 22 | 0 | 0 | 0 | 0 | 0 | 0 | no | NCC -ve |
| 23 | 0 | 0 | 0 | 0 | 0 | 0 | no | NCC -ve |
| 24 | 0 | 0 | 0 | 0 | 0 | 0 | no | NCC -ve |
| 25 | 14 | 0 | 3 | 0 | 15 | 2 | active | NCC +ve |
| 26 | 0 | 0 | 0 | 0 | 0 | 0 | no | NCC -ve |
| 27 | 0 | 0 | 0 | 0 | 0 | 0 | no | NCC -ve |
| 28 | 0 | 0 | 0 | 0 | 0 | 0 | no | NCC -ve |
| 29 | 0 | 0 | 0 | 0 | 0 | 0 | no | NCC -ve |
| 30 | 0 | 0 | 0 | 0 | 0 | 0 | no | NCC -ve |
| 31 | 0 | 0 | 0 | 0 | 0 | 0 | no | NCC -ve |
| 32 | 0 | 0 | 0 | 0 | 0 | 0 | no | NCC -ve |
| 33 | 0 | 0 | 0 | 0 | 0 | 0 | no | NCC -ve |
| 34 | 0 | 0 | 0 | 0 | 0 | 0 | no | NCC -ve |
| 35 | 0 | 0 | 0 | 0 | 0 | 0 | no | NCC -ve |
| 36 | 0 | 0 | 0 | 0 | 0 | 0 | no | NCC -ve |

ID = serial identification number of patients: + ve = positive; -ve = negative; NCC = neurocysticercosis

CC was detected in 7 (90%) of the 8 samples. An LR+ of 2.23 suggests that individuals with cysticercosis are approximately 2.23 times more likely to have a positive test result than those without cysticercosis. An LR- of 0.21 suggests that individuals without cysticercosis were approximately 0.21 times as likely to have a positive test result compared to their counterparts with the condition.

### 4.5. Positive predictive value and negative predictive value for serum cysticercosis tests

The positive predictive value (PPV) is estimated at 38.9% with a 95% confidence interval ranging from 23.0 to 54.8, suggesting that among individuals testing positive for cysticercosis, approximately more than a quarter are correctly diagnosed. While the negative predictive value (NPV) is estimated at 94.4% with a 95% confidence interval spanning from 87.0 to 101.9. This indicates that among individuals testing negative for cysticercosis, the likelihood of them truly being negative is quite high (Table 5).

### 4.6. Receiver operating characteristics curve

The ROC curve's summary statistic, or area under the curve (AUC) value, indicates how well the test can differentiate between those who are diseased and those who are not diseased. In our case, the area under the ROC curve of 0.78 indicates that the test has a good ability to distinguish between neurocysticercosis positive and negative (Fig 2).

## 5.0. Discussion

Findings in this study show that; the appropriate validity of the serum cysticercosis tests was achieved by parallel testing of the serum cysticercosis tests. The multiple evaluation methods including sensitivity and specificity, likelihood ratios, positive and negative predictive values, and receiver operating characteristics curve provided different insights into the effectiveness of the test.

### 5.1. Parallel testing of serum cysticercosis tests

It is possible to run several diagnostic tests consecutively or simultaneously in clinical settings. In consecutive testing, the tests are performed in sequential order i.e. starting with one test and then followed by others. In this manner of testing, a test is only considered to be positive if all tests give positive results and in any case of discordant results, normally a third tie-breaker is needed. When assessing a single diagnostic test, doctors, or clinicians, can either manage the condition empirically without testing or conduct the test and then connect the test findings with the chosen course of action.

When testing a patient in parallel, two or more tests are administered simultaneously. The patient is regarded as positive for the tested disease if any test results show up as positive and this is what was done in our study. Having two serological cysticercosis tests, particularly when conducted simultaneously, really increased our confidence in selecting suitable patients for the next steps which were neuroimaging and treatment decision-making. For our environment, the study sites were far from the point of analysis of the serum tests, the next step of CT scan examinations was costly, and the final desired step of treatment had its complications.

Parallel testing raises sensitivity and lowers false negatives; hence it improves the negative predictive value. On the other hand, sequential testing raises specificity as a result of reducing false positives ending up improving positive predictive value.

The decision to carry out parallel or serial testing of the diagnostic tests depends on the currently existing environment, the type of disease to be diagnosed, and the target outcomes whether increased sensitivity or specificity. A study in Brazil by Arruda et al., carried out both methods of parallel and serial methods in studying Sensitivity and specificity for the detection of canine *Leishmania* infection. They obtained higher sensitivity in parallel testing and higher specificity in serial testing of the same diagnostic tests [27].

Table 3.  Distribution of serum cysticercosis tests and neuroimaging (Computed Tomography).

| Characteristic | Overall | Kongwa | Chunya | p-value** |
|---|---|---|---|---|
| | n (%) | n (%) | n (%) | |
| **Serum cysticercosis test(s)** | **N = 223** | **N = 172** | **N = 51** | |
| **CC Ag (apDia Ag test)** | | | | **<0.001** |
| Negative | 208 (93.3) | 166 (96.5) | 42 (82.4) | |
| Positive | 15 (6.7) | 6 (3.5) | 9 (17.6) | |
| **CC Ab (LDBio IgG test)** | | | | >0.9 |
| Negative | 206 (92.4) | 159 (92.4) | 47 (92.2) | |
| Positive | 17 (7.6) | 13 (7.6) | 4 (7.8) | |
| **Cysticercosis** | | | | **0.030** |
| Negative | 198 (88.8) | 157 (91.3) | 41 (80.4) | |
| Positive (on either test) | 25 (11.2) | 15 (8.7) | 10 (19.6) | |
| **Cerebral computed tomography** | **N = 36** | **N = 26** | **N = 10** | |
| **Neurocysticercosis** | | | | **0.024** |
| NCC -ve | 28 (77.8) | 23 (88.5) | 5 (50.0) | |
| NCC +ve | 8 (22.2) | 3 (11.5) | 5 (50.0) | |

** Fisher's exact test; Pearson's Chi-squared test

Table 4.  Serum cysticercosis test results vs computed tomography for neurocysticercosis (N = 36).

| Test | NCC results | | Total |
|---|---|---|---|
| **CC** | **Positive** | **Negative** | |
| | n (%) | n (%) | n |
| Positive | 7 (87.5) | 11(39.3) | 18 |
| Negative | 1(12.5) | 17(60.7) | 18 |
| **Total** | 8 | 28 | 36 |

Table 5.  Positive predictive value and negative predictive value for serum cysticercosis tests.

| CC diagnostic test | Point estimate (%) | Interval estimate (95%CI) |
|---|---|---|
| PPV | 38.9 | 23.0–54.8 |
| NPV | 94.4 | 87.0–101.9 |

In a remote community in Zambia, Mubanga et al. and Tanzania, Stelzle et al., evaluated the diagnostic accuracy of a point-of-care test that detects antibodies to diagnose *Taenia solium* cysticercosis [28,29]. Three tests i.e. the LLGP EITB (Sensitivity 98%, Specificity 100%), the rT24H EITB (Sensitivity 94%, Specificity 98%), and the B158/60 serum Ag ELISA (Sensitivity 80%, Specificity 97%) were used as reference tests. The two components of reference tests i.e. EITB tests were performed in parallel to raise the sensitivity. Different sample sizes among these two studies yielded different outcomes whereby a sample size of 255 in Zambia and 601 in Tanzania resulted in the Sensitivity of 35% (95% CI: 14–63%) and 49% (uncertainty interval [UI] 41–58), respectively. Both cases didn't perform well, something which called for possibly titrating antigen and other reagents' concentration in the strip to produce a performance similar to existing cysticercosis tests such as the rT24H EITB [28].

We assumed that by administering two serological tests we can synergize the probability of picking up the correct candidates for the next step of CT scan. The CT scan examination

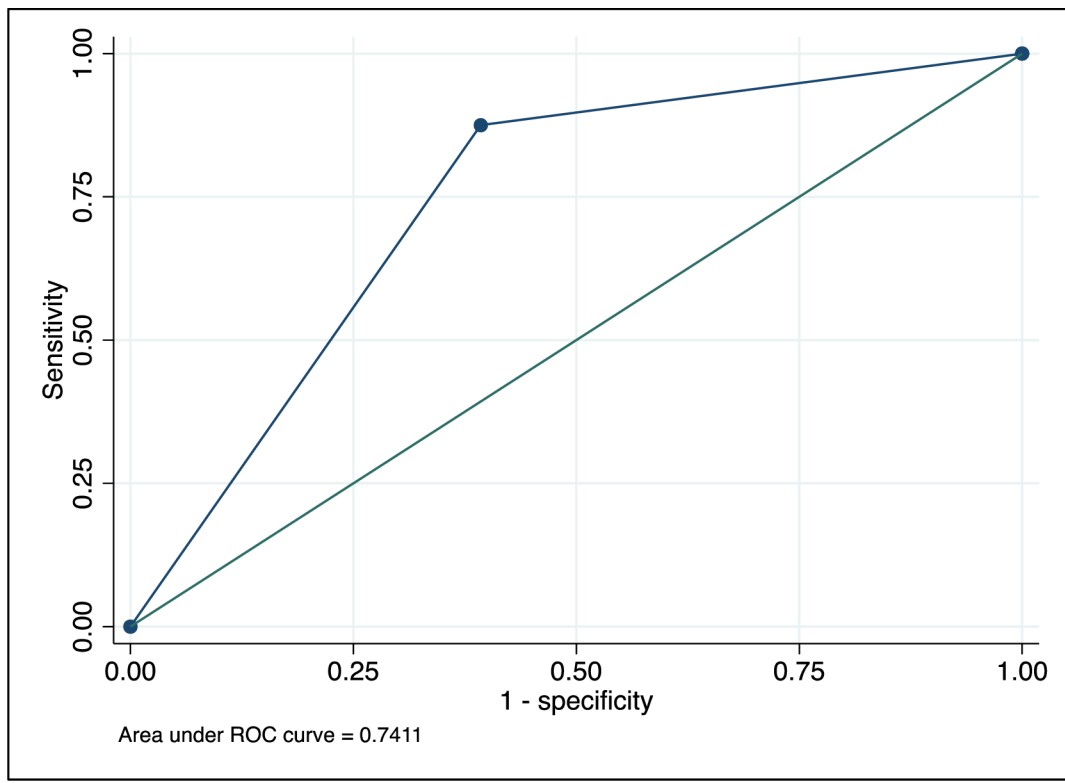

**Fig 2. The receiver operating characteristic curve.**

was hardly accessible and costly during the time of our study conduct. The treatment decision needed a panel decision of experts depending on the size, number, and location of lesions.

The two tests were utilized to assess a single serological status because, based on the information at hand, sufficient confidence was required to proceed with the cerebral CT scan evaluation. As a result, we opted to process the data from both tests in parallel, regardless of the data from each of the tests.

## 5.2. Sensitivity and specificity of the cysticercosis serological tests

While the sensitivity and specificity values given by the manufacturer offer significant information about the test's performance under ideal circumstances, local context reviews enable us to grasp its actual diagnostic value. Population variances, disease prevalence, as well as the methods used to carry out the test might all greatly affect its validity.

Our study demonstrated a higher sensitivity which means the test was reliable for the screening as it detects most of the diseased individuals. The moderate specificity is technically suggestive for the confirmatory tests for the positive results. Therefore, we can conclude from these results that the serum cysticercosis tests can do better in screening the NCC due to its higher sensitivity, but it should not be relied on, that's, needs a two step-approach being followed by a confirmatory test for the definitive diagnosis because of its moderate specificity.

Santini et al. (2021) emphasized the need to evaluate how tests perform in real-world conditions showing that biases and variations often have an impact on test accuracy. They talked about the calculation and explanation of key metrics including sensitivity, specificity, Positive Predictive Value (PPV), Negative Predictive Value (NPV), and the Receiver Operating Characteristic (ROC) curve. The test's ability to diagnose a target condition relies heavily on

these elements [30]. Santini et al. also pointed out that these factors are more than just numbers; they have real-world effects on how we assess and interpret diagnostic accuracy when we think about how common a disease is and what testing methods we use [30].

Therefore, although our study results match Santini et al.'s including the obtained sensitivity and specificity of the serological test, it is essential to realize these values have to be interpreted in the setting of the population and nature factors under which the test is used. Moreover, our results on the ROC and AUC provide more evidence of the accuracy of the test in this particular environment.

## 5.3. Likelihood ratio of the cysticercosis serological tests

An alternative metric for summarizing diagnostic accuracy is a likelihood ratio. Although they are not often utilized, their potent qualities make them more beneficial in therapeutic settings. The likelihood ratio provides a summary of the number of times that patients with the condition are more (or less) likely than patients without the disease to have that specific outcome. Formally speaking, it is the ratio of the likelihood of a particular test result in those with the illness to the likelihood in those without. A higher probability ratio than one signifies a correlation between the test result and the existence of the disease, whereas a lower likelihood ratio suggests a correlation between the test result and the absence of the condition. The greater the evidence is for the presence or absence of illness; the farther the likelihood ratios are from the value of one. A likelihood ratio was used in the diagnostic accuracy study by da Silva-Etto et al evaluating two rapid stool antigen tests using an immunochromatographic assay to detect *Helicobacter pylori*; and one of them performed better than the other in clinical practice [31]. Scientific evidence presented by Yougui Wu suggests that population-based paired design yields the best likelihood ratio for the accuracy of diagnostic tests [32]. Although we conducted a cross-sectional study, it paves a foundation for the future robust diagnostic accuracy test evaluation since it gives a broad understanding of the test's performance and initial evaluation.

## 5.4. Predictive values and receiver operating characteristics (ROC) curve

The terms "positive predictive value" (PPV) and "negative predictive value" (NPV) refer to the likelihood that a person with a positive test result has the condition, whilst the latter shows the likelihood that a person with a negative test result does not. The prevalence of the investigated disease or condition in the community being examined has an impact on these values in addition to the accuracy of the test. Santini et al established the understanding of the basic study design for diagnostic accuracy and therefore used different methods including predictive values and receiver operating characteristics [30]. Our current study suggests a mixed performance of low PPV and higher NPV. The low PPV might reflect a low disease prevalence in our study population. The high NPV suggests that individuals testing negative are truly disease-free which is a good indicator of screening ability to rule out cysticercosis. So long as we target early detection of the disease and our goal is screening, the test has done well. However, in the case of the diagnosis, additional confirmatory tests like imaging are needed.

The receiver operating characteristic curve visually illustrates the test's ability to discriminate between individuals with and without the condition. The curve illustrates the trade-off between sensitivity and specificity across various test result levels [33].

The area under the curve -ROC in our current study suggested moderate to good diagnostic performance. This means that the test can reasonably distinguish between NCC-positive and NCC-negative individuals, although for more accuracy, an additional confirmatory test may be needed.

Having seen such results we can build a case that several existing conditions including population diversity in the sense that the population in which the current testing is performed may be different from the manufacturer's tested population in terms of genetic diversity and the burden of the disease [34,35]. Policy-makers need enough evidence before they allow the new diagnostic tests to be used in their local settings or commit a dedicated budget to purchase most of the new technologies.

## 6.0. Strength and limitation

### 6.1. Strength

People with epilepsy (PWE) in our environment were able to be tested and detected by the cysticercosis serological tests. Performing a local evaluation normally allows for sustainable ongoing quality control and assurance which can help to identify gaps between the manufacturer's specification and real-life performance. Although we admit the value of neuroimaging as well as serological cysticercosis tests in the diagnostic process, we are aware that the use of neuroimaging as a strength in our research needs clarification. Only one of the eight subjects who tested NCC positive for a CT scan came from the group of cysticercosis serological-negative individuals. Seven of them were positive for both serum cysticercosis and a CT scan. This limited number of neuroimaging assessments means we cannot fully compare the serological test results with neuroimaging findings across all positive cases. However, still including neuroimaging in at least one situation offers useful information for the diagnosis process and useful knowledge on cysticercosis in this community.

To enable a more thorough analysis of the link between serological findings and imaging results, hence fortifying the conclusions reached from such a comparison; we recommend future studies to offer neuroimaging for every serology-positive subject.

### 6.2 Limitation

The duration of the research conduct was short and resulted in a limited number of participants. Otherwise, the sensitivity and specificity can be influenced by biological sample handling i.e., sample collection, shipping, and storage depending on the existing environment [36]. The whole sample handling has been explained in the methodology and therefore, our environment might influence the validity of the outcomes in one way or another. This follows a long distance between the site of sample collection to the point of analysis i.e., the minimum distance was about 70 km (Chunya -NIMR-Mbeya laboratory) and the maximum was 675 km (Kongwa -NIMR-Mbeya laboratory). This needs careful handling of the samples including local sample collection, storage, shipping, and analysis. It's a heavy task under routine practice and a small fault at any point might affect the validity of the tests. Otherwise, the tests are sophisticated, expensive, only commercially available in our settings and their equipment is not portable. Therefore, it needs higher investment in its infrastructure.

But again, during the conduct of this study, CT scan services were not locally available in the settings of the recruitment sites. Participants had to travel long distance to obtain these services as mentioned in the methodology.

Both cysticercosis serological tests and cerebral CT scans are not affordable among people with epilepsy (PWE), they are also hardly accessible and need a lot of investment by the existing local authorities.

## 7.0. Conclusion

This research emphasized the considerable impact of neurocysticercosis (NCC) on individuals with epilepsy (PWE) in the Kongwa and Chunya districts, where 11.2% of participants

were found to be positive for cysticercosis, and 22.2% of those who underwent neuroimaging exhibited lesions characteristic of NCC. The sensitivity and specificity of 87.5% and 60.7% respectively for the serum cysticercosis tests predict its usefulness, although there are limitations in identifying all cases that were confirmed by the computed tomography. Furthermore, the positive predictive value and negative predictive value of 38.9% and 94.4% respectively demonstrate both strength and constraints in the low resource countries like Tanzania.

The multiple diagnostic evaluation techniques, such as sensitivity, specificity, predictive values, likelihood ratios, and receiver operating characteristics curve (ROC), are crucial for verifying the tests' ability to offer distinct perspectives on the test's efficacy. This robustness of the approach increases confidence in the use of serum cysticercosis tests in our routine practice, even though sensitivity and specificity may have been impacted by our biological sample processing. In every other case, PWEs were able to access diagnostic health services which were locally not available since their facilities were not providing such services before. Overall, this study provides essential evidence that can inform practical application as well as policy changes, marking a step forward in the field of *T. solium* neurocysticercosis diagnostic services. All in all, this study demonstrated that parallel testing of the serum cysticercosis tests produced the proper validity of the tests.

## Acknowledgments

We express our gratitude to Kongwa District Hospital's Mr. Masura Nyakarungu, Dr. Nazar Temba, Mr. Augustino Seganje, and Sr. Valentine Mbula for their committed support in making this work successful. We are grateful to Dr. Daniel Msungu, Mr. Isaka Kyando, and the whole CYSTINET-Africa personnel in Chunya district hospital for their unwavering work, support, and help in fieldwork, including data collecting and laboratory work for this study,

## Author contributions

**Conceptualization:** Charles Elias Makasi.

**Formal analysis:** Charles Elias Makasi, Godfrey Guga, Michael J. Mahande, Johnson Mshiu.

**Investigation:** Charles Elias Makasi, Bernard Ngowi.

**Methodology:** Charles Elias Makasi, Bernard Ngowi, Abisai Kisinda.

**Project administration:** Bernard Ngowi.

**Supervision:** Bernard Ngowi, Andrew Kilale, Michael J. Mahande, Blindina T Mmbaga.

**Validation:** Bernard Ngowi, Andrew Kilale, Michael J. Mahande, Blindina T Mmbaga.

**Visualization:** Bernard Ngowi, Andrew Kilale, Godfrey Guga, Michael J. Mahande, Johnson Mshiu, Blindina T Mmbaga.

**Writing – original draft:** Charles Elias Makasi.

**Writing – review & editing:** Charles Elias Makasi, Bernard Ngowi, Andrew Kilale, Godfrey Guga, Michael J. Mahande, Johnson Mshiu, Abisai Kisinda, Blindina T Mmbaga.

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
