## [Decision Letter · Decision Letter 0]

20 Oct 2024

Dear Dr Makasi,

Thank you very much for submitting your manuscript "Correlation of serum screening and neuroimaging studies: a comparative analysis for diagnostic evaluation of neurocysticercosis among people with epilepsy attending mental health clinics in selected district hospitals of Tanzania" for consideration at PLOS Neglected Tropical Diseases. As with all papers reviewed by the journal, your manuscript was reviewed by members of the editorial board and by several independent reviewers. In light of the reviews (below this email), we would like to invite the resubmission of a significantly-revised version that takes into account the reviewers' comments. 

We cannot make any decision about publication until we have seen the revised manuscript and your response to the reviewers' comments. Your revised manuscript is also likely to be sent to reviewers for further evaluation.

Sincerely,

Feng Xue, Ph.D.

Guest Editor

Jong-Yil Chai

Section Editor

Reviewer's Responses to Questions

**Key Review Criteria Required for Acceptance?**

**Methods**

-Are the objectives of the study clearly articulated with a clear testable hypothesis stated?

-Is the study design appropriate to address the stated objectives?

-Is the population clearly described and appropriate for the hypothesis being tested?

-Is the sample size sufficient to ensure adequate power to address the hypothesis being tested?

-Were correct statistical analysis used to support conclusions?

-Are there concerns about ethical or regulatory requirements being met?

Reviewer #1: The aim of the study has been mentioned , however author could specify the objectives of the study clearly aligned to conclusion. Although its a part of a larger study, the author could write about the study design sample size calculation and and sampling method in detail. The statistical methods used were appropriate and ethical concerns were addressed, however the author could share the sample of participation information sheet to bring more clarity.

Reviewer #2: The methodology seems appropriate and well described, with most procedures clearly outlined.

However, some parts seem a bit confusing to me, due to repeated information. In line 226, it mentions the sample collection locations, providing the same information again in line 230

Reviewer #3: L94-95: Rephrase and emphasize on blood collection, volume, anticoagulant…. 

The material and methods, result and discussion sections need to be shorten, please be more specific, avoid too much details. If methods have been already published use citation. For commercial kits use “following manufacturers procedure”.

**Results**

-Does the analysis presented match the analysis plan?

-Are the results clearly and completely presented?

-Are the figures (Tables, Images) of sufficient quality for clarity?

Reviewer #1: The analysis plan is appropriate, however, figure 1 and table 1 data is missing from result section. Likewise Figure 2 and table-4 data is not matching.(It was mentioned 8 (22.2%) of 36 PWE who went for CT scan; (7 from the CC-positive group and one from PWE from CC negative) with neurological presentation showed NCC-typical lesions, But in Table 4 it was 10 with NCC). Author could correct the numbers where ever necessary .

Reviewer #2: The results, although interesting, seem a little confusing to me in the way they are described. There is a lot of information that seems unnecessary to me, because it is in the table (i.e. p-values), and even highlighted in bold. 

It would be interesting to show an image of the CT scan

Reviewer #3: The material and methods, result and discussion sections need to be shorten, please be more specific, avoid too much details. If methods have been already published use citation. For commercial kits use “following manufacturers procedure”.

**Conclusions**

-Are the conclusions supported by the data presented?

-Are the limitations of analysis clearly described?

-Do the authors discuss how these data can be helpful to advance our understanding of the topic under study?

-Is public health relevance addressed?

Reviewer #1: The author could rephrase the conclusion aligned to the aim, i.e: to assess the correlation between serum screening and neuroimaging studies among PWE attending mental health clinics in selected district hospitals of Tanzania.

All other aspects has been mentioned clearly.

Reviewer #2: The results are well discussed, in a well-organized way, and supported by a good list of bibliographic references, leading to a supported conclusion. It is worth highlighting the fact that it presents the strengths and limitations of the study, which I think is an advantage of the document.

Reviewer #3: Conclusion need to be improved.

**Editorial and Data Presentation Modifications?**

Reviewer #1: Minor Revision

Reviewer #2: Lines 101, 103, 104,105, 113, 114, 234 – (T.solium) should add space after “T.” - T. solium

Line 103 – missed the "a" at “cellulosae and ”italics – Cysticercus cellulosae 

Line 232 – Should be T. solium 

Line 549 - missed the italics – Helicobacter pylori

Lines 533, 538, 554, 559, 572, 597, 606, 619, 640, 645 – missed the italics – Taenia solium

Line 636 - missed the italics – Leishmania

Line 653 - missed the italics – Helicobacter pylori

Figure 1 - should review how to cite the source of the image

Reviewer #3: (No Response)

**Summary and General Comments**

Reviewer #1: Congratulations to the author for choosing a pertinent topic for research. Over all the quality of the article is satisfactory.

Reviewer #2: This paper is about the correlation of serum screening and neuroimaging in the diagnostic evaluation of neurocysticercosis among people with epilepsy attending mental health clinics in Tanzania. 

The objectives are well-defined, and the inclusion criteria for the study are well-described. The authors present a good flowchart of the recruitment of people for the study, respecting ethical principles. 

The subject of the paper is very relevant due to the complexity of the diagnostics of cysticercosis and the relevance of the disease, that can lead to epilepsy. 

As I mentioned, the article is well written and can be easier to read and understand, with some improvements in some confusing parts, especially in the results section.

Reviewer #3: In Taenia solium endemic communities with limited hygienic infrastructure, epilepsy associated neurocysticercosis become to be a major a public health problem that require improvement in diagnostic capacities and intervention strategies. People living in contact with subjects infected with Taenia Solium are at higher risk to acquire parasite eggs through the fecal-oral route that eventually become larvae in several tissues including the central nervous system. While larvae become calcified in the tissues, several neurological complications are produced, including epilepsy. Neuroimaging is the most reliable diagnostic tool, but access is limited by high cost where it is more needed. Alternatively, several efforts have been done to develop sensitive and specific serological tools (ELISA and Westernblot) to increase diagnostic capacities in endemic areas. The manuscript by Charles E. Makasia and coworkers is following the serological approach and described the correlation between a serological test for neurocysticercosis and neuroimaging studies. 

L144: PWE is already defined “L 116” not need to repeat.

L159: “consecutively selected” please replace with “enrolled prospectively”

L160: consider “with or without”… anti-seizure medication

L169 – 190: Consider shortening, too much detail. Epilepsia definition should be included in the inclusion criteria. Avoid text repetition. 

This reviewers consider this is an important contribution, but extensive description dilute the main findings.

PLOS authors have the option to publish the peer review history of their article (what does this mean? ). If published, this will include your full peer review and any attached files.

**Do you want your identity to be public for this peer review?** For information about this choice, including consent withdrawal, please see our Privacy Policy .

Reviewer #1: Yes: Dr Jarina begum, Professor in department of Community Medicine, Manipal Tata Medical College, Jamshedpur, Jharkhand, India

Reviewer #2: No

Reviewer #3: No
---

## [Decision Letter · Decision Letter 1]

15 Jan 2025

PNTD-D-24-01083R1

Correlation of serum screening and neuroimaging studies: a comparative analysis for diagnostic evaluation of neurocysticercosis among people with epilepsy attending mental health clinics in selected district hospitals of Tanzania

Dear Dr. Makasi,

Thank you for submitting your manuscript to PLOS Neglected Tropical Diseases. After careful consideration, we feel that it has merit but does not fully meet PLOS Neglected Tropical Diseases's publication criteria as it currently stands. Therefore, we invite you to submit a revised version of the manuscript that addresses the points raised during the review process.

Please submit your revised manuscript within 60 days Feb 14 2025 11:59PM. If you will need more time than this to complete your revisions, please reply to this message or contact the journal office at plosntds@plos.org. Please include the following items when submitting your revised manuscript:

We look forward to receiving your revised manuscript.

Kind regards,

Feng Xue, Ph.D.

Guest Editor

Jong-Yil Chai

Section Editor

Shaden Kamhawi

co-Editor-in-Chief

Paul Brindley

co-Editor-in-Chief

**Journal Requirements:**

At this stage, the following Authors/Authors require contributions: Charles Elias Makasi, Bernard Ngowi, Andrew Kilale, Godfrey Guga, Michael Mahande, Johnson Mshiu, Abisai Kisinda, and Blindina T Mmbaga. Please ensure that the full contributions of each author are acknowledged in the "Add/Edit/Remove Authors" section of our submission form.

2) Please ensure that the funders and grant numbers match between the Financial Disclosure field and the Funding Information tab in your submission form. Note that the funders must be provided in the same order in both places as well.

**Reviewers' Comments:**

Reviewer's Responses to Questions

**Key Review Criteria Required for Acceptance?**

**Methods**

-Are the objectives of the study clearly articulated with a clear testable hypothesis stated?

-Is the study design appropriate to address the stated objectives?

-Is the population clearly described and appropriate for the hypothesis being tested?

-Is the sample size sufficient to ensure adequate power to address the hypothesis being tested?

-Were correct statistical analysis used to support conclusions?

-Are there concerns about ethical or regulatory requirements being met?

Reviewer #1: The revised format has incorporated all suggestions. However, the objectives mentioned by the author as a reply could be written after mentioning the aim of the study at the end of Introduction section of the article. Rest all looks good to go. All the best or final proofreading and publication.

Reviewer #2: The suggested modifications have been made, upgrading the paper. The methodology is now more clear.

Reviewer #4: -Are the objectives of the study clearly articulated with a clear testable hypothesis stated? YES

-Is the study design appropriate to address the stated objectives? YES

-Is the population clearly described and appropriate for the hypothesis being tested? YES

-Is the sample size sufficient to ensure adequate power to address the hypothesis being tested? NO

-Were correct statistical analysis used to support conclusions? YES

-Are there concerns about ethical or regulatory requirements being met? NO

**Results**

-Does the analysis presented match the analysis plan?

-Are the results clearly and completely presented?

-Are the figures (Tables, Images) of sufficient quality for clarity?

Reviewer #1: Yes, the suggested corrections have been done by the author.

Reviewer #2: The suggested modifications have been made. The results are more clear and easier to understand.

Reviewer #4: -Does the analysis presented match the analysis plan? YES

-Are the results clearly and completely presented? YES

-Are the figures (Tables, Images) of sufficient quality for clarity? NO

**Conclusions**

-Are the conclusions supported by the data presented?

-Are the limitations of analysis clearly described?

-Do the authors discuss how these data can be helpful to advance our understanding of the topic under study?

-Is public health relevance addressed?

Reviewer #1: Yes. The author has mentioned conclusion appropriately along with addressing the public health relevance in the revised document.

Reviewer #2: The conclusions were improved as was suggested by the reviewers, and are now aligned with the objectives of the work

Reviewer #4: -Are the conclusions supported by the data presented? NO

-Are the limitations of analysis clearly described? YES

-Do the authors discuss how these data can be helpful to advance our understanding of the topic under study? YES

-Is public health relevance addressed? YES

**Editorial and Data Presentation Modifications?**

Reviewer #1: Minor revision: Adding the objectives of the study at the end of introduction in the main article file

Reviewer #2: Despite the corrections made, there are some small changes that need to be made

lines210, 211 - Should be "ml", instead of "mls"

Lines 217, 232 - should be "ºC", instead of "degree centigrade"

Line 318 - should be "p=0.030", not p<0.03 (according the table 3)

Line 320 - should be "p=0.024", not p<0.024 (according the table 3)

Lines 553, 557,571,575,586,609,617,627,631,648,652,657,665 - You missed the italics

Reviewer #4: The authors have done great work in assessing the "Correlation of serum screening and neuroimaging studies: a comparative analysis for diagnostic evaluation of neurocysticercosis among people with epilepsy attending mental health clinics in selected district hospitals of Tanzania"

The manuscript has the ability to contribute to the scientific body of knowledge regarding decisions making on who should be taken for neuroimaging in T. solium endemic areas. However, the manuscript has a number of typos and areas requiring clarifications. Please refer to the following comments and suggestions

INTRODUCTION

Line 102: Kindly replace the word “Guts” with intestines as guts is more colloquial

Line 104: The placing of porcine cysticercosis in brackets at the end of the sentence is misleading as it now means that T. solium larval stage is called porcine cysticercosis.

Line 113: The phrase T. solium-caused conditions should be replaced with T. solium associated conditions. This is because T. solium per say does not cause epilepsy.

Line 121-123: The techniques for diagnosis of NCC are available (as explained in line 126) as they have been published. However, its the availability, knowledge and lack of equipment to perform the tests that is challenging. The authors should consider rephrasing the sentence and combine with the follow up statement in line 124-126.

Line 134: Remove the words “But also” and start as a new paragraph.

METHODS

Line 144: Indicate early here when the study was conducted and the dates can then be removed from line 165 and don’t need to be mentioned again in line 177.

Line 146: Remove the word respectively and indicate the regions for the two study sites e.g Kongwa district hospital in Dodoma, central Tanzania and Chunya hospital in Mbeya, Southern Tanzania.

Line 149: Only mention Makasi et., al. once and enter the citations e.g. as described by Makasi et al (21,22)

Line 169: The authors should define the abbreviation “CT” and thereafter use the abbreviation CT scan

Line 176: The authors do not need to state that CM was an employee of the study. Also check the sentence “attended potential participants” and instead write “attended to potential participants”

PROCEDURES:

Overall, this paragraph needs to be restructured. There is no need to mention each district hospital separately with the specific days when the mental health clinics are held (unless this has an effect on the results). In addition, it has already been mentioned in the study design and setting that the study was done in Kongwa and Chunya district hospitals. The authors should focus on outlining the procedures done during the study such as described from line 186. I find the included information from line 181 to part of 186 unnecessary for the procedures section.

Line 194: What do the authors mean by “convenience was applied”?

Line 198: Add “s” to participant. Also replace “whereby” with “and” specify if the written information is written informed consent signed

Line 200: The authors refer to collection of biological samples. What are these other biological samples collected in addition to stool and blood samples? In any case this section is about data collection methods and tools therefore, collection of samples should move downwards to Test methods

Line 201: Rephrase the sentence to read “After sample collection, further data was collected from participants by CM and SN, the trained medical practitioners with neurological experience.”

The mentioning of DS (which does not appear anywhere else in the manuscript) and CYSTINET-Africa is not necessary here.

Line 207: What do the authors mean by “and neurological status by the use of in-depth

neurological and neurological examination standardized and validated forms”? This statement is confusing

Line 212-214: This statement indicates that the Red top bottles were left at room temperature for one hour and serum was obtained. However, the subsequent sentence (line 214) indicates that the Red top containers were left in the fridge and serum only obtained after centrifugation. The authors should phrase these sentences correctly to avoid confusion of the process.

Line 216: The table may be removed if the authors just state that serum was collected in three cryovials. One for testing with apDia Ag-ELISA, the second for LDBio IgG Western blot and the third for quality control.

Line 217: kindly complete the sentence “similar barcodes/unique…..?”

Line 229: Change the phrase “certain amount of sample”… to “certain number of samples” If the agreement was to send samples when they were 25 or more, the authors should state so without creating ambiguity like a certain number of samples.

Line 230-232: Kindly include appropriate punctuations to make the sentence easy to understand. If the professional laboratory experts are the trained laboratory staff then state as such

Line 234: Replace the sentence “with an escort of the laboratory expert” with “accompanied by a trained laboratory staff”

Line 236: replace “done” with “performed” and “procedures” with “guidelines”

Line 241: Replace “every” with “all” and “for one” with “on either”

Line 242-243: state that the CT scan examination was performed as described by Makasi et al 2024. and no need to highlight it in board colour

Line 243: Remove “The” from the beginning of the sentence.

Line 244: Remove “The” before NCC

Line 247: The authors to explain what CR represents. In addition, the authors use so many abbreviations for names of persons and these abbreviations are not stated anywhere. The authors should just state what was done without the trouble of indicating exactly the names of people who performed certain functions.

Line 249: The authors have already stated that the revised 2017 Del Brutto criteria was used to make NCC diagnosis. This is a repetition.

Line 251: What does (source) represent?

Line 252-253: The authors should simply state that treatment options for patients with NCC was agreed upon by the study clinical team which also included a neurologist. The abbreviated names will appear in the acknowledgments.

Line 256: The authors should state what treatment was given to the patients with NCC

Line 283: The abbreviations Ab, Ag have not been defined anywhere before

RESULTS

Line 283: Delete the sentence “i.e kongwa and Chunya”

Line 286: The authors should begin by presenting first the demographic data before presenting the results of the serological tests and CT scan examination.

Line 286-287: the sentence “neuroimaging (cerebral computed tomography (CT) scan examination)” should read (cerebral CT scanning). Provided the CT abbreviation is explained as suggested for line 169 above

Fig 1: A box is missing to explain those CC-ve without neurological signs/symptoms (somewhere between N=196 and N=23)

304-305: Replace “pattern in” with “pattern in terms of”.. and place a full stop instead of a coma after participants.

305: Replace the sentence “more than twice PWE” to “more than twice the number of PWE”

306: Replace the sentence “twice as many as farmers” to “twice the proportion of farmers”

Line 317 and Table 3 cysticercosis positive: The authors should indicate that the 25 CC+ were positive on either test.

Line 320: delete the repetition of “i.e Kongwa; 13. (7.6%) vs Chunya 4 (7.8%) This already mentioned above.

330-331: The authors should check the phrasing of the sentence especially the last part of the sentence.

Table 4: The results of the table can be incorporated into Table 3 under Neurocysticercosis. This way the 7 and 1 CC positive participants would be known in terms of the hospital they came from.

Table 5: This is not necessary as it is simply explained in the text

Line 350; check the value for the PPV

DISCUSSION

Line 390: Consider replacing the sentence “was very dangerous.” The authors may consider using “had its own complications”

Line 404: remove “…for infection” after the specificity in the brackets

**Summary and General Comments**

Reviewer #1: Congratulations to the authors for addressing the all the queries and incorporating all the suggestions while revising the article file. Best wishes for final publication.

Reviewer #2: As i mentioned before, the objectives of the work are well-defined, and the

inclusion criteria for the study are well described. The authors present a good

flowchart of the recruitment of people for the

study, respecting ethical principles.

The subject of the paper is very relevant due to

the complexity of the diagnostics of

cysticercosis and the relevance of the disease,

that can lead to epilepsy. All the modifications made in the manuscript improved the article, made it easier to read and more complete.

Reviewer #4: 1. 286-297: The authors should have a logical flow of the results. Start with how many from your total 223 were CC positive as in line 284. Of these, how many also had neurological signs/symptoms? Next how many of the 223 were CC negative and of these, how many had neurological signs/symptoms. Next how many of the CC positive had a CT scan examination and the reason for those missing. Then how many of the CC negative with neurological signs/symptoms had a CT scan examination and how many missed? Finally, state how many of those who had a CT scan examination had NCC lesions (both from the CC+ and from the CC- with Neurological signs/symptoms.

The authors should avoid repeating PWE after every number as they have already stated that they recruited PWE from the mental health clinic.

2. The study aims at determining the Correlation of serum screening and neuroimaging studies for diagnostic evaluation of neurocysticercosis. However, the authors have not described the NCC lesions that were seen from neuroimaging examination for an informed decision regarding correlation with serum screening. This description would enhance our understanding of the correlation

3. Line 328: The authors state in the methods that diagnosis of NCC would be based on the revised Del Brutto 2017 criteria and would define probable and definite NCC. However, here they refer to a Gold standard test without mentioning the Del Brutto Criteria

4. The sample size for the cysticercosis positive participants cannot be used to make a meaningful conclusion regarding the sensitivity, specificity, PPV, NPV of the serological tests in relation to neuroimaging.in fact the authors do not state what the estimated sample size was. The authors should consider providing this information.

5.Line 420-432: The authors have not discussed sensitivity and specificity in the context of their findings. For example, the cited paper of Santini et., al clearly discussed their findings. It would greatly improve our understanding of the two parameters if discussed within context of the authors findings/results. The same applies to the likelihood ratio. The authors have at least made reference to their findings in the discussion of PPV/NPV and ROC

6. Line 475-488: The authors correctly state the multiple analysis used as a strength of their study. However, the fact that participants were tested with cysticercosis tests and that neuroimaging was performed cannot be considered as strengths unless they state that all CC positives had Neurimaging to be used for their comparative analysis

PLOS authors have the option to publish the peer review history of their article (what does this mean? ). If published, this will include your full peer review and any attached files.

**Do you want your identity to be public for this peer review?** For information about this choice, including consent withdrawal, please see our Privacy Policy .

Reviewer #1: **Yes: ** Dr Jarina Begum, Professor in Department of Community Medicine, Manipal Tata Medical College, Jamshedpur, Manipal Academy of Higher education, Manipal, India.

Reviewer #2: No

Reviewer #4: No

**Figure resubmission:**
---

## [Decision Letter · Decision Letter 2]

18 Feb 2025

PNTD-D-24-01083R2Correlation of serum screening and neuroimaging studies: a comparative analysis for diagnostic evaluation of neurocysticercosis among people with epilepsy attending mental health clinics in selected district hospitals of TanzaniaPLOS Neglected Tropical DiseasesDear Dr. Makasi, Thank you for submitting your manuscript to PLOS Neglected Tropical Diseases. After careful consideration, we feel that it has merit but does not fully meet PLOS Neglected Tropical Diseases's publication criteria as it currently stands. Therefore, we invite you to submit a revised version of the manuscript that addresses the points raised during the review process. Please submit your revised manuscript within 30 days Mar 20 2025 11:59PM. If you will need more time than this to complete your revisions, please reply to this message or contact the journal office at plosntds@plos.org. Please include the following items when submitting your revised manuscript: * A rebuttal letter that responds to each point raised by the editor and reviewer(s). You should upload this letter as a separate file labeled 'Response to Reviewers '. This file does not need to include responses to any formatting updates and technical items listed in the 'Journal Requirements' section below. * A marked-up copy of your manuscript that highlights changes made to the original version. You should upload this as a separate file labeled 'Revised Manuscript with Track Changes '. * An unmarked version of your revised paper without tracked changes. You should upload this as a separate file labeled 'Manuscript '. If you would like to make changes to your financial disclosure, competing interests statement, or data availability statement, please make these updates within the submission form at the time of resubmission. Guidelines for resubmitting your figure files are available below the reviewer comments at the end of this letter. We look forward to receiving your revised manuscript. Kind regards, Feng Xue, Ph.D.Guest EditorPLOS Neglected Tropical Diseases Jong-Yil ChaiSection EditorPLOS Neglected Tropical Diseases

Shaden Kamhawi

co-Editor-in-Chief

Paul Brindley

co-Editor-in-Chief

**Journal Requirements:**

Please ensure that the funders and grant numbers match between the Financial Disclosure field and the Funding Information tab in your submission form. Note that the funders must be provided in the same order in both places as well.

State the initials, alongside each funding source, of each author to receive each grant. For example: "This work was supported by the National Institutes of Health (####### to AM; ###### to CJ) and the National Science Foundation (###### to AM).".

**Reviewers' comments:** Reviewer's Responses to Questions

**Key Review Criteria Required for Acceptance?**

**Methods** :

-Are the objectives of the study clearly articulated with a clear testable hypothesis stated?

-Is the study design appropriate to address the stated objectives?

-Is the population clearly described and appropriate for the hypothesis being tested?

-Is the sample size sufficient to ensure adequate power to address the hypothesis being tested?

-Were correct statistical analysis used to support conclusions?

-Are there concerns about ethical or regulatory requirements being met?

Reviewer #1: The revised version is clear & concise with all suggested modifications.

Reviewer #2: Are the objectives of the study clearly articulated with a clear testable hypothesis stated? Yes

-Is the study design appropriate to address the stated objectives? Yes

-Is the population clearly described and appropriate for the hypothesis being tested? Yes

-Is the sample size sufficient to ensure adequate power to address the hypothesis being tested? Yes

-Were correct statistical analysis used to support conclusions? Yes

-Are there concerns about ethical or regulatory requirements being met? No

Reviewer #4: -Are the objectives of the study clearly articulated with a clear testable hypothesis stated? YES

-Is the study design appropriate to address the stated objectives? YES

-Is the population clearly described and appropriate for the hypothesis being tested? YES

-Is the sample size sufficient to ensure adequate power to address the hypothesis being tested? YES

-Were correct statistical analysis used to support conclusions? YES

-Are there concerns about ethical or regulatory requirements being met? NO

**Results** :

-Does the analysis presented match the analysis plan?

-Are the results clearly and completely presented?

-Are the figures (Tables, Images) of sufficient quality for clarity?

Reviewer #1: Yes results are complete with clear presentation.

Reviewer #2: Does the analysis presented match the analysis plan? Yes

-Are the results clearly and completely presented? Yes

-Are the figures (Tables, Images) of sufficient quality for clarity? Yes

Reviewer #4: -Does the analysis presented match the analysis plan? YES

-Are the results clearly and completely presented? YES

-Are the figures (Tables, Images) of sufficient quality for clarity? YES

**Conclusions** :

-Are the conclusions supported by the data presented?

-Are the limitations of analysis clearly described?

-Do the authors discuss how these data can be helpful to advance our understanding of the topic under study?

-Is public health relevance addressed?

Reviewer #1: Conclusion along with strengths and limitations described appropriately. Public health relevance has been addressed aswell.

Reviewer #2: Are the conclusions supported by the data presented? Yes

-Are the limitations of analysis clearly described? Yes

-Do the authors discuss how these data can be helpful to advance our understanding of the topic under study? Yes

-Is public health relevance addressed? Yes

Reviewer #4: -Are the conclusions supported by the data presented? YES

-Are the limitations of analysis clearly described? YES

-Do the authors discuss how these data can be helpful to advance our understanding of the topic under study? NO

-Is public health relevance addressed? YES

A few general comments were not addressed by the authors. I will repeat them here in order to enhance our understanding of their findings:

1. The study aims at determining the Correlation of serum screening and neuroimaging studies for diagnostic evaluation of neurocysticercosis. However, the authors have not described the NCC lesions that were seen from neuroimaging examination for an informed decision regarding correlation with serum screening. This description would enhance our understanding of the correlation

2. Line 310: The authors stated in the methods that diagnosis of NCC would be based on the revised Del Brutto 2017 criteria and would define probable and definite NCC. However, there is no mention anywhere in the results about this but here they refer to a Gold standard test without mentioning the Del Brutto 2017 Criteria. The authors can shed more light on this.

3.Line 402-414: The authors have not discussed sensitivity and specificity in the context of their findings. For example, the cited paper of Santini et., al. clearly discussed their findings. It would greatly improve our understanding of the two parameters if discussed within the context of the authors findings/results. The same applies to the likelihood ratio. The authors have at least made reference to their findings in the discussion of PPV/NPV and ROC

4. Line 475-488: The authors correctly state the multiple analysis used as a strength of their study. However, the fact that participants were tested with cysticercosis tests and that neuroimaging was performed cannot be considered as strengths unless they state that all CC positives had Neuroimaging to be used for their comparative analysis. An explanation for this would help understand their reasoning.

**Editorial and Data Presentation Modifications?**

Reviewer #1: Nil

Reviewer #2: Despite the corrections made, there are some small changes that still need to be made

Lines 656, 673 - missed the italics (Leismania; Helicobacter pylori)

Reviewer #4: Minor

Line 165: The authors should define here the abbreviation "CT" e.g Computed tomography (CT) scans. After this the authors can now just use "CT scan" without writing it in full.

line 269: As suggested earlier the (cerebral computed tomography (CT) scan examination) can be written simply as (cerebral CT scan examination) if the CT abbreviation is defined earlier as suggested above.

Line 288: Consider replacing the sentence ".....more than twice PWE without..." to "more than twice the number of PWE without..."

Line 312-312: The authors should check this sentence. Add % to (87.5). what does the last part mean (...and 1 (12.5%) case which could not be picked serum cysticercosis test)???

Line 328: the PPV is written as 38.9.0% is this the correct nomenclature or the authors meant 38.9%?

**Summary and General Comments** :

Reviewer #1: The article is of public health importance. However, future research is recommended with a larger sample for generalizability.

Reviewer #2: (No Response)

Reviewer #4: The authors have done great work in revising the manuscript and it now reads well. However, the responses to the above stated comments especially how they link their results in the discussion section of the paper would help in arriving at a decision for the manuscript.

PLOS authors have the option to publish the peer review history of their article (what does this mean? ). If published, this will include your full peer review and any attached files.

**Do you want your identity to be public for this peer review?** For information about this choice, including consent withdrawal, please see our Privacy Policy .

Reviewer #1: **Yes: ** Jarina Begum

Reviewer #2: No

Reviewer #4: No

---

## [Decision Letter · Decision Letter 3]

20 Mar 2025

Dear Dr Makasi,

We are pleased to inform you that your manuscript 'Correlation of serum screening and neuroimaging studies: a comparative analysis for diagnostic evaluation of neurocysticercosis among people with epilepsy attending mental health clinics in selected district hospitals of Tanzania' has been provisionally accepted for publication in PLOS Neglected Tropical Diseases.

Best regards,

Feng Xue, Ph.D.

Guest Editor

Jong-Yil Chai

Section Editor

Shaden Kamhawi

co-Editor-in-Chief

Paul Brindley

co-Editor-in-Chief

Reviewer's Responses to Questions

**Key Review Criteria Required for Acceptance?**

**Methods**

-Are the objectives of the study clearly articulated with a clear testable hypothesis stated?

-Is the study design appropriate to address the stated objectives?

-Is the population clearly described and appropriate for the hypothesis being tested?

-Is the sample size sufficient to ensure adequate power to address the hypothesis being tested?

-Were correct statistical analysis used to support conclusions?

-Are there concerns about ethical or regulatory requirements being met?

Reviewer #1: Yes, the objectives are aligned with methodology and all requirements being met.

Reviewer #2: Are the objectives of the study clearly articulated with a clear testable hypothesis stated? Yes

-Is the study design appropriate to address the stated objectives? Yes

-Is the population clearly described and appropriate for the hypothesis being tested? Yes

-Is the sample size sufficient to ensure adequate power to address the hypothesis being tested? Yes

-Were correct statistical analysis used to support conclusions? Yes

-Are there concerns about ethical or regulatory requirements being met? No

Reviewer #4: -Are the objectives of the study clearly articulated with a clear testable hypothesis stated? YES

-Is the study design appropriate to address the stated objectives? YES

-Is the population clearly described and appropriate for the hypothesis being tested? YES

-Is the sample size sufficient to ensure adequate power to address the hypothesis being tested? YES

-Were correct statistical analysis used to support conclusions? YES

-Are there concerns about ethical or regulatory requirements being met? YES

**Results**

-Does the analysis presented match the analysis plan?

-Are the results clearly and completely presented?

-Are the figures (Tables, Images) of sufficient quality for clarity?

Reviewer #1: Yes, the results are presented appropriately.

Reviewer #2: -Does the analysis presented match the analysis plan? Yes

-Are the results clearly and completely presented? Yes

-Are the figures (Tables, Images) of sufficient quality for clarity? Yes

Reviewer #4: -Does the analysis presented match the analysis plan? YES

-Are the results clearly and completely presented? YES

-Are the figures (Tables, Images) of sufficient quality for clarity? YES

**Conclusions**

-Are the conclusions supported by the data presented?

-Are the limitations of analysis clearly described?

-Do the authors discuss how these data can be helpful to advance our understanding of the topic under study?

-Is public health relevance addressed?

Reviewer #1: Conclusions are aligned to objectives and limitations are described

Reviewer #2: Are the conclusions supported by the data presented? Yes

-Are the limitations of analysis clearly described? Yes

-Do the authors discuss how these data can be helpful to advance our understanding of the topic under study? Yes

-Is public health relevance addressed? Yes

Reviewer #4: -Are the conclusions supported by the data presented? YES

-Are the limitations of analysis clearly described? YES

-Do the authors discuss how these data can be helpful to advance our understanding of the topic under study? YES

-Is public health relevance addressed? YES

**Editorial and Data Presentation Modifications?**

Reviewer #1: Accept

Reviewer #2: (No Response)

Reviewer #4: Accept the manuscript

**Summary and General Comments**

Reviewer #1: Overall, the research has addressed pertinent public health problem.

Reviewer #2: I think that the modifications suggested by the reviewers were made, improving the manuscript. The papers is now more clear and easier to understand.

Reviewer #4: The authors have addressed the concerns raised

PLOS authors have the option to publish the peer review history of their article (what does this mean? ). If published, this will include your full peer review and any attached files.

**Do you want your identity to be public for this peer review?** For information about this choice, including consent withdrawal, please see our Privacy Policy .

Reviewer #1: **Yes: ** Jarina Begum

Reviewer #2: No

Reviewer #4: No

---

## [Editor Report · Acceptance letter]

Dear Dr Makasi,

We are delighted to inform you that your manuscript, "Correlation of serum screening and neuroimaging studies: a comparative analysis for diagnostic evaluation of neurocysticercosis among people with epilepsy attending mental health clinics in selected district hospitals of Tanzania," has been formally accepted for publication in PLOS Neglected Tropical Diseases.

Best regards,

Shaden Kamhawi

co-Editor-in-Chief

Paul Brindley

co-Editor-in-Chief
